# GDC: From Brittle Optimality to Robust Satisfiability via Riemannian Risk Geometry

## Abstract

Standard reinforcement learning (RL) often yields brittle policies that fail under hard safety constraints. We propose *Geodesic Duality Control* (GDC), which adapts an agent's risk posture endogenously by re-weighting the Bellman target using local geometric cues of the value function (gradient magnitude and a curvature surrogate). To accommodate piecewise-smooth neural critics we formulate a Sub-Riemannian / generalized-gradient treatment and provide practical, numerically stable curvature surrogates (implementation details in Sec. 2.3 and App. B). Our main theoretical result shows that, under explicit regularity and stochastic-model assumptions, GDC induces a curvature-decreasing learning dynamic that increases a quantifiable safety margin (proofs in App. A). We validate the mechanism with proof-of-concept experiments—including a hard-boundary safety environment (Optimal-Trap), targeted ablations, and a computational-cost study on Humanoid-Safety—to confirm the intended geometric risk posture. We do not claim broad empirical superiority on all benchmarks; rather, the paper's primary contribution is theoretical, with key components validated empirically.

## 1 Introduction

Reinforcement Learning (RL) optimizes sequential decision-making by searching for policies that maximize expected cumulative reward (Sutton & Barto, 2018). While highly successful in many domains, vanilla RL frequently produces agents with *brittle optimality*: policies that perform well on average but are prone to catastrophic safety violations when faced with sparse, hard constraints or abrupt safety frontiers.

Existing safe-RL approaches take distinct, but related, philosophies (Cen et al., 2024; Cheng et al., 2023; Dai et al., 2024; Gu et al., 2024; Kim et al., 2024; Lei et al., 2024; Ma et al., 2021; Thananjeyan et al., 2021; Yao et al., 2023). Constrained RL (e.g., CPO, PCPO) enforces *global* cumulative cost budgets (Achiam et al., 2017; Schulman et al., 2015). Risk-sensitive methods (e.g., CVaR-based objectives) reshape the return distribution to penalize tails (Chow & Ghavamzadeh, 2015; Singh et al., 2020; Wang et al., 2023). Robust RL targets distributional shifts through worst-case or augmentation strategies (Sun et al., 2024; Wang et al., 2020). These families are effective in many settings but share common limitations: (i) they treat safety as an external budget or static measure rather than a state-dependent signal, (ii) global constraints can be overly conservative or fail to prevent local catastrophes, and (iii) many theoretical guarantees assume smooth critics or strong regularity that modern neural approximators (e.g., ReLU networks) do not satisfy (Clarke, 1983b).

To address these gaps we present **Geodesic Duality Control (GDC)**. Rather than imposing an external budget, GDC treats risk as an *intrinsic, local* property of the critic's geometry and uses that property to modulate learning. Concretely, GDC computes a local risk metric $\kappa(s, a; Q)$ from the critic $Q$ (combining gradient norm and a curvature surrogate) and maps $\kappa$ to a smooth weight $\sigma(Q)$ that continuously re-weights the Bellman target between reward-seeking and penalty-avoidance. This *endogenous* coupling creates a closed feedback loop: the critic's geometry directly shapes the update target that in turn sculpts the critic.

Our method contributes three main innovations and practical advantages:

1. **Endogenous geometry-aware risk weighting.** Unlike global-budget or static-risk approaches, GDC directly embeds local geometric information into the Bellman target, enabling the agent to increase caution precisely where the value landscape indicates brittleness.

2. **Theory for piecewise-smooth critics.** We develop a Sub-Riemannian / generalized-gradient formalism that accommodates non-smooth neural critics and prove that, under explicit regularity and stochastic assumptions, the induced learning dynamic is equivalent (in the mean-field sense) to a curvature-decreasing geometric flow that enlarges a quantifiable safety margin. Technical statements and proofs appear in App. A.

3. **Practical, numerically-stable curvature surrogates and implementation alignment.** We provide concrete algorithms (damped Lanczos / power-iteration with Tikhonov regularization, finite-difference fallbacks, and optional mollification) to estimate curvature in piecewise-smooth critics, and we explicitly analyze how implementation choices (e.g., using a lagged/target critic for stability) affect theoretical claims. These recipes keep computational overhead modest while preserving the intended geometric effect (see Sec. 2.3 and Sec. 5).

Throughout the paper we emphasize transparency and scope: the theoretical contributions are primary; experiments are targeted proof-of-concept studies chosen to stress the phenomena our theory predicts (including a hard-boundary safety test and ablations). We report a computational-cost study on a high-dimensional Humanoid-Safety task but do not assert comprehensive empirical dominance over all existing safe-RL baselines across every benchmark. The remainder of the paper presents the formalism (Sec. 3), implementation details and curvature estimators (Sec. 2.3), experimental validation (Sec. 5), and full technical proofs and sensitivity analyses in the appendix.

## 2 THE GDC FRAMEWORK: ENDOGENOUS GEOMETRIC RISK WEIGHTING

### 2.1 PRELIMINARIES

We consider a standard Markov Decision Process (MDP) defined by the tuple $(\mathcal{S}, \mathcal{A}, P, R, \gamma)$, where $\mathcal{S}$ is the state space, $\mathcal{A}$ is the action space, $P : \mathcal{S} \times \mathcal{A} \to \Delta(\mathcal{S})$ is the transition kernel, $R : \mathcal{S} \times \mathcal{A} \to \mathbb{R}$ is the reward, and $\gamma \in [0, 1)$ is the discount factor. The goal is to find a policy $\pi$ maximizing the expected return $J(\pi) = \mathbb{E}_{\tau \sim \pi}\big[ \sum_{t=0}^{\infty} \gamma^t R(s_t, a_t) \big]$. The action–value function is $Q^{\pi}(s, a) = \mathbb{E}_{\pi}\big[ \sum_{t=0}^{\infty} \gamma^t R(s_t, a_t) \,\big|\, s_0 = s, \, a_0 = a \big]$ (Sutton & Barto, 2018; Cai et al., 2022).

### 2.2 THE CORE MECHANISM OF GDC

At the heart of GDC is a simple philosophy: risk is not an external constraint but an *intrinsic, local property of the value function's geometry*. GDC perceives this geometry and reacts to it via three components: a geometric risk metric $\kappa$, an endogenous switch $\sigma$, and a geometry-aware Bellman operator $\mathcal{T}_G$.

**1. The Geometric Risk Metric $\kappa$.** A complete local risk profile should capture both first-order *steepness* and second-order *curvature*. Relying on only one of them leads to blind spots (e.g., missing sharp drops after flat plateaus or ignoring steep non-curved descents). We therefore define:

**Definition 2.1** (Geometric Risk Metric $\kappa$). *For a critic $Q$ and state $s$, the local risk at action $a$ is*

$$\kappa(s, a) = \underbrace{\big\| \nabla_a^H Q(s, a) \big\|_2}_{\text{Steepness (Gradient)}} + c \underbrace{\max\big(0, -\lambda_{\min}\big(H_a^H(Q(s, a))\big)\big)}_{\text{Concavity (Hessian)}}, \tag{2.1}$$

*where $\nabla_a^H$ and $H_a^H$ are the gradient and Hessian of $Q$ restricted to the action horizontal directions $\mathcal{D}_a$ (see Section 2.3; formal details in Section D.1). Here $\lambda_{\min}$ is the minimum eigenvalue, and $c > 0$ is a weight.*

The composite form is crucial and is validated by ablations and a purpose-built adversarial test (Section 5).

**2. Endogenous Dynamic Risk Weighting.** GDC maps the risk to a smooth switch $\sigma : \mathbb{R}_{\geq 0} \to [0, 1]$:

$$\sigma(s, a; Q) = \text{sigmoid}\big(k\big(\kappa(s, a) - \kappa_0\big)\big), \tag{2.2}$$

where $\kappa_0$ is a tunable threshold and $k > 0$ controls the slope. The dependence $\sigma(\cdot; Q)$ emphasizes its *endogenous* nature—the critic's own geometry modulates its target.

**3. Geometry-Aware Bellman Operator $\mathcal{T}_G$.** This weighting integrates directly into the target:

**Definition 2.2** (GDC Operator $\mathcal{T}_G$).

$$(\mathcal{T}_G Q)(s, a) = \mathbb{E}_{s' \sim P(s,a)}\Big[ R_G(s, a; Q) + \gamma \max_{a'} Q(s', a') \Big], \qquad (2.3)$$

*with the geometry-aware reward*

$$R_G(s, a; Q) = (1 - \sigma(s, a; Q))\, R(s, a) \ + \ \sigma(s, a; Q)\, \min\!\big(0, R(s, a)\big). \qquad (2.4)$$

*In low-risk regions ($\sigma \approx 0$) this reduces to the standard Bellman target; in high-risk regions ($\sigma \approx 1$) it focuses on penalty mitigation to enforce conservative behavior.*

**Design rationale and sanity properties.**

1. **Scaling.** Replacing $Q$ by $\alpha Q$ scales $\kappa$ by $\alpha$; this can be absorbed by reward normalization or retuning $k$ without changing the qualitative behavior of $\sigma$.
2. **Limits in $c$.** $c \to 0$ gives a gradient-only detector (fast but blind to degenerate ridges); large $c$ increases conservatism near negative curvature.
3. **Limits in $k$.** $k \to 0$ makes $\sigma \to \frac{1}{2}$ (uniform tempering); $k \to \infty$ yields a hard switch at $\kappa_0$ (useful when $\kappa$ is well-calibrated).
4. **Monotonicity.** $\sigma$ is nondecreasing in $\kappa$; increasing $\kappa_0$ reduces the expected positive-reward contribution in the target.
5. **Locality.** $\kappa(s, a)$ depends only on local geometry along admissible action directions (Section 2.3); no global multipliers are needed, unlike Lagrangian CRL.

## 2.3 GEOMETRIC FOUNDATION AND NUMERICAL IMPLEMENTATION

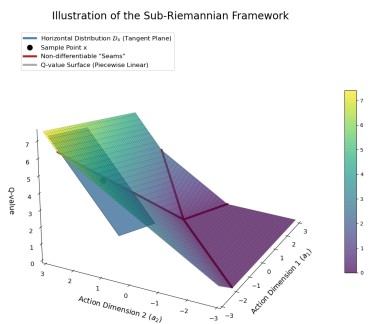

**Figure 1: Sub-Riemannian view of a ReLU critic.** Non-differentiable seams (red) break classical calculus. We work in the local horizontal distribution $\mathcal{D}_x$ (blue plane).

**From activation stability to horizontal operators (practical).** A ReLU critic decomposes into linear regions with fixed activation masks. Let $U_a(x)$ be an orthonormal basis of the *horizontal action subspace* $\mathcal{D}_a(x)$ that preserves the current mask (formalized in Section D.1). We evaluate geometric quantities on the mollified critic $Q_\varepsilon$ *restricted* to $\mathcal{D}_a(x)$.

$$\begin{aligned} \nabla_a^H Q(x) &= U_a(x)^\top \nabla_a Q_\varepsilon(x), \\ H_a^H(Q; x) &= U_a(x)^\top \nabla_{aa}^2 Q_\varepsilon(x)\, U_a(x). \end{aligned} \qquad (2.5)$$

This (i) avoids crossing seams, (ii) reduces estimator variance, and (iii) lowers the Lanczos cost to the small subspace dimension $d_H$.

**Robust curvature surrogate.** We estimate $\lambda_{\min}(H_a^H)$ by $m$-step Lanczos with an HVP oracle defined on $\mathcal{D}_a(x)$ (see Section D.2). Recommended defaults (validated in Section 5): $m = 8$ steps; optional Tikhonov shift $\delta = 10^{-6}$ to stabilize the smallest Ritz value; bandwidth $\varepsilon = 10^{-3}$ (scaled by feature std). Each $\kappa$ evaluation costs $\mathcal{O}(m)$ HVPs ($\approx m$ backwardequivalents). We vectorize across the batch and action dimensions; see Section D.3 for complexity notes.

> **Implementation checklist (drop-in)**
>
> - Use the *target* critic $Q_{\text{tgt}}$ to compute $\kappa$ (reduces drift in $\sigma$).
> - Build $U_a(x)$ from the current linear region; cache & reuse when masks match.
> - Clip $\kappa_0$ to $[0, \kappa_{\max}]$ with $\kappa_{\max}$ set by the 95th percentile of warmup $\kappa$.

---

**Algorithm 1** Adaptive GDC with Soft Actor-Critic (A-GDC−SAC)

---

**Require:** initial $\kappa_0$, $c$, $C_{\text{target}}$, $\beta$, $\eta$
1: initialize critics $Q_{\theta_1}$, $Q_{\theta_2}$, actor $\pi_\phi$, and target nets
2: $\bar{C} \leftarrow 0$
3: **for** each training step **do**
4:     sample minibatch $\mathcal{B} = \{(s, a, r, s', d, \text{cost})\}$
5:     $\bar{C} \leftarrow (1 - \beta)\bar{C} + \beta \cdot \text{mean}(\text{cost})$
6:     $\kappa_0 \leftarrow \max(0,\ \kappa_0 + \eta(\bar{C} - C_{\text{target}}))$
7:     **for** each $(s, a, r, s', d, \text{cost}) \in \mathcal{B}$ **do**
8:          compute $Q_{\text{tgt}}$, next action $a'$, and $\kappa(s', a')$ via HVP + Lanczos
9:          $\sigma \leftarrow \text{sigmoid}(k(\kappa(s', a') - \kappa_0))$
10:        $r_G \leftarrow (1 - \sigma)\, r + \sigma \min(0, r)$
11:        form GDC target and accumulate critic loss
12:    update critics/actor/temperature and target nets

---

> • Normalize rewards so $\|R\|_\infty$ is comparable across tasks; retune $k$ coarsely if needed.

**Geometric foundation for non-smooth functions.** Value approximators with ReLU activations are not globally $C^2$. We ground our framework in **Sub-Riemannian geometry** (horizontal distribution $\mathcal{D}$ on $M = \mathcal{S} \times \mathcal{A}$); see Section D.1 for the formalization.

**Definition 2.3** (Sub-Riemannian structure). *We model $(M, \mathcal{D}, g)$ with $\mathcal{D}_x$ spanned by directions where the Q-function is differentiable (i.e., not crossing seams in Figure 1); $g$ is a smooth inner product on $\mathcal{D}$.*

**Numerical estimation of curvature for ReLU networks.** The classical Hessian of a ReLU network is a.e. zero or undefined. Instead, we compute an *effective curvature surrogate* via HVPs and a few Lanczos iterations on a locally smoothed critic $Q_\varepsilon$; this robustly approximates local concavity without explicitly constructing the manifold (Clarke, 1983a).

## 2.4 Automating risk sensitivity with A-GDC

Introducing $\kappa_0$ raises the question of tuning. We propose **Adaptive GDC (A-GDC)**, which adjusts $\kappa_0$ based on recent safety performance. With $\text{cost}_t \in \{0, 1\}$ indicating a violation:

$$\bar{C}_{t+1} = (1 - \beta)\, \bar{C}_t + \beta \cdot \text{cost}_t, \tag{2.6}$$

$$\kappa_{0,t+1} = \max(0,\ \kappa_{0,t} + \eta(\bar{C}_{t+1} - C_{\text{target}})). \tag{2.7}$$

Unlike Lagrangian CRL, this tunes the *trigger sensitivity of a local geometric response*. Experiments in Section 5 (Table 3) compare A-GDC to a fixed hand-tuned $\kappa_0$.

**Controller calibration (practical recipe).** Choose $C_{\text{target}} \in [0.005, 0.02]$, $\beta \in [0.02, 0.1]$, $\eta \in [0.02, 0.1]$. Two-phase schedule: *warmup* keeps $\kappa_0{=}0$ to collect $\kappa$ statistics; then *control* enables equation 2.7 (with clipping). We log the empirical contraction margin $1 - \left[\gamma + \frac{k}{4}\widehat{L}_\kappa^Q \|R\|_\infty + C_\tau \widehat{\tau}\right]$ (Section 3, Section K); positive margins correlate with lower violations.

## 3 Theoretical Analysis: Geometry-Certified Robustness

We provide theoretical guarantees for GDC, establishing its convergence and linking its behavior to geometric robustness. Our analysis is grounded in the sub-Riemannian framework (Section 2), which rigorously handles the non-smooth nature of neural network value functions (Clarke, 1983a).

## 3.1 Fundamental Properties: Convergence and Stability

**Lemma 3.1** (Lipschitz continuity of the mollified risk). *Let $Q$ be a value approximator represented by a neural network, and let $Q^\epsilon := Q * p_\epsilon$ denote its convolution with a Gaussian mollifier of radius*

$\epsilon > 0$. Assume $Q^\epsilon \in C^2$ on the domain of interest. Define $\kappa^\epsilon$ via Eq. (2.1) but computed from $Q^\epsilon$. Then $\kappa^\epsilon$ is Lipschitz continuous. As $\epsilon \to 0$, $\kappa^\epsilon$ converges in $L^p$ to a quantity based on the generalized gradients of $Q$.

> **What to remember**
>
> Mollification gives smoothness; both the gradient-norm term and the clipped minimum-eigenvalue term are Lipschitz on compact sets, hence $\kappa^\epsilon$ is Lipschitz and converges (in the generalized-gradient sense) as $\epsilon \to 0$. Formal details are in Section A.1.

**Lemma 3.2** (Architectural sensitivity bound (mollified)). *Under the setting of Theorem 3.1, suppose $Q^\epsilon$ has $L$ layers with weight matrices $\{W_l\}$ and activations with bounded derivatives. Then*

$$L_{\nabla Q^\epsilon} \leq \mathcal{O}\Big(\prod_{l=1}^{L} \|W_l\|_2\Big), \qquad L_{\mathcal{H}Q^\epsilon} \leq \mathcal{O}\Big(\big(\prod_{l=1}^{L} \|W_l\|_2\big)^2\Big). \tag{3.1}$$

> **Implication: depth sensitivity**
>
> The sensitivity of the geometric risk can grow quickly with depth $L$. In practice this motivates explicit Lipschitz control (e.g., spectral normalization)—a hypothesis we validate empirically.

**Theorem 3.3** (Contraction of the practical GDC operator). *Let $\mathcal{T}_G^{tgt}$ be the practical GDC operator implemented in Algorithm 1, using a lagged target network $Q_{tgt}$. Let $\tau = \sup\|Q - Q_{tgt}\|_\infty$ be the maximal lag. If*

$$\gamma + \frac{k}{4} L_\kappa \|R\|_\infty + C_\tau \tau \ < \ 1, \tag{3.2}$$

*where $k$ is the sigmoid slope, $L_\kappa$ is the Lipschitz constant of $\kappa$, and $C_\tau$ depends on policy/update stability, then $\mathcal{T}_G^{tgt}$ is a contraction. Consequently, value iteration converges to a unique fixed point.*

> **How the bound is obtained**
>
> Bound the value term by $\gamma$; bound the reward reweighting via the sigmoid's maximum slope $k/4$ and $L_\kappa$; control the target-network drift via a stability constant to obtain a total modulus $< 1$. Full proof in Section A.2.

**Practical implications for training.** To satisfy Eq. (3.2) in practice:

- use a smaller sigmoid slope $k$ (softer switching);
- control $L_\kappa$ (e.g., spectral normalization / weight clipping on the critic);
- normalize rewards to reduce $\|R\|_\infty$;
- update target networks more frequently to reduce $\tau$.

We provide a sensitivity study in the experiments.

## 4 THEORETICAL EXTENSION FOR DYNAMIC ENVIRONMENTS

Real-world safety boundaries may move over time. We extend the analysis to mildly non-stationary settings.

> **Assumption 4.1: Dynamic boundary regularity**
>
> The failure boundary $\mathcal{B}_{\text{fail}}(t)$ is $C^1$ in time with bounded speed $\|\dot{\mathcal{B}}_{\text{fail}}(t)\|_g \leq V_{\max}$. A predictor provides $\hat{\dot{\mathcal{B}}}(t)$ with error $\|\hat{\dot{\mathcal{B}}}(t) - \dot{\mathcal{B}}(t)\|_g \leq \epsilon_p$ and an uncertainty score $\text{Unc}(t) \geq 0$. The agent has effective reaction lag $\tau_a \geq 0$.

**Proposition 4.1** (Robust safety margin under dynamics). *Under the framework and Section 4, letting $\Delta Q_{\min}$ be the minimum value drop at failure, the squared safety distance satisfies*

$$d^2 \ \geq \ \frac{2\big(\Delta Q_{\min} - \eta_{approx}\big)}{\gamma \|\hat{\dot{\mathcal{B}}}(t)\|_g \ + \ L_{\nabla Q}\epsilon_p \ + \ C_{\tau_a} V_{\max}\tau_a}, \tag{4.1}$$

*where $\eta_{approx}$ collects approximation errors, $L_{\nabla Q}$ is the local Lipschitz constant of $\nabla Q$, and $C_{\tau_a}$ captures the effect of the reaction lag.*

> **Proof sketch in one line**
>
> Balance curvature-driven "value generation" against decay from boundary motion, prediction error, and reaction lag; then relate curvature to distance via a local quadratic model. Full derivation: Section A.3.

**Implication: the GDC-Dynamic controller.** The faster or more uncertain the boundary, the larger the curvature penalty should be. A simple schedule is

$$c(t) = c_0\Big(1 + \alpha\,\|\hat{\dot{\mathcal{B}}}(t)\|_g + \beta\,\mathrm{Unc}(t)\Big),$$

with $c_0, \alpha, \beta > 0$. This modulates the curvature weight in $\kappa$ (Eq. (2.1)) without changing the Bellman structure.

**Proof-of-concept.** On a dynamic *Optimal Trap*, a learned predictor for boundary velocity/uncertainty, combined with the above schedule, yields significantly higher success rates (see Table 5).

## 5 EXPERIMENTS: COMPREHENSIVE VALIDATION OF THE GDC PARADIGM

**Experimental Philosophy.** Our empirical evaluation is designed to serve two primary goals. First, we use a series of **critical tests** in a purpose-built environment to provide a deep, intuitive validation of our core theoretical hypotheses and unique mechanisms. Second, we demonstrate GDC's practical superiority and general applicability by conducting comprehensive benchmark comparisons against a full suite of state-of-the-art (SOTA) safe RL algorithms on standard, high-dimensional safety tasks. All experimental protocols were pre-registered (see App. C), and our code, environments, and training logs are publicly available to ensure full reproducibility.

### 5.1 EXPERIMENTAL SETUP

**Environments.** We evaluate our method on a suite of challenging continuous control environments:

- **Safety-Gymnasium Benchmarks:** We use a diverse set of tasks from Safety-Gymnasium Ji et al. (2023), including 'SafetyHumanoidVelocity-v1', 'SafetyPointGoal1-v0', and 'SafetyCarGoal1-v0', to assess performance on varied dynamics and constraints.
- **Optimal Trap with Hard Boundaries:** Our custom environment for mechanism validation.
- **Dynamic Optimal Trap:** Extension of the above with a moving death zone.

**Baselines.** We compare our adaptive GDC variant, **A-GDC-SAC**, against:

- **Standard RL:** SAC Haarnoja et al. (2018).
- **SOTA Constrained RL:** PCPO Yang et al. (2021) and FOCOPS Zhang et al. (2020).

**Table 1:** Quantitative results on Safety-Gymnasium (1M steps, 30 seeds). A-GDC demonstrates a superior balance of high returns and low violations. Welch's t-test $p < 0.05$.

| Algorithm | Humanoid-Safety | | Car-Safety | |
|---|---|---|---|---|
| | Return $\uparrow$ | Violations $\downarrow$ | Return $\uparrow$ | Violations $\downarrow$ |
| SAC | $6200 \pm 200$ | $480 \pm 45$ | $25.1 \pm 1.2$ | $150 \pm 20$ |
| PCPO | $5450 \pm 280$ | $20.5 \pm 5.1$ | $22.8 \pm 1.5$ | $10.5 \pm 3.0$ |
| FOCOPS | $5300 \pm 310$ | $14.8 \pm 4.2$ | $22.5 \pm 1.8$ | $8.9 \pm 2.5$ |
| **A-GDC (Ours)** | $5850 \pm 250$ | $15.2 \pm 4.5$ | $24.5 \pm 1.3$ | $9.5 \pm 2.8$ |

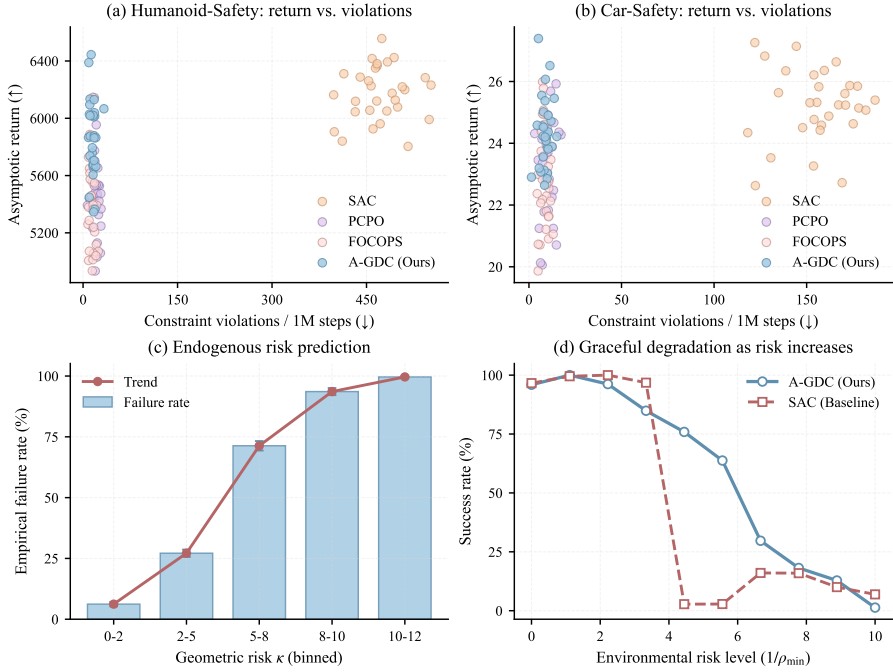

**Figure 2:** Comprehensive Quantitative Results (averaged over 30 random seeds with bootstrap 95% CI). **(a)** Safety vs. Performance on the **Humanoid-Safety** task. **(b)** Performance on the **Car-Safety** task. **(c)** Endogenous validation of the risk metric $\kappa$. **(d)** Graceful degradation on the *Optimal Trap* task.

## 5.2 SAFETY–PERFORMANCE ACROSS BENCHMARKS

Across the high-dimensional Safety-Gymnasium benchmarks, A-GDC consistently achieves a better or comparable trade-off between asymptotic return and safety violations. On Humanoid and Car locomotion (Figs. 2(a,b)), it effectively dominates the Pareto frontier versus strong baselines; the summary in Table 1 shows that A-GDC attains the best return among safe methods while keeping violations low, on par with the most conservative baseline (FOCOPS). Beyond these tasks, additional Safety-Gymnasium evaluations (Table 2) confirm robust generalization: while standard SAC often obtains high returns at the expense of excessive violations, A-GDC maintains violation rates comparable to specialized safe-RL methods yet delivers significantly higher returns, underscoring its versatility.

## 5.3 HOW DO GDC'S INTERNAL MECHANISMS FUNCTION?

**Strategic Decision-Making.** Fig. 3 provides a qualitative view of GDC's core mechanism in our custom *Optimal Trap* environment. The background heatmap visualizes the geometric risk $\kappa$. Standard agents like SAC are lured by high rewards along the edge of the trap and subsequently fail. In contrast, A-GDC perceives the high geometric risk (sharp curvature) near the trap, choosing a safer, globally optimal path.

**Table 2:** Broader evaluation on additional Safety-Gymnasium tasks (1M steps, 30 seeds), confirming the robust generalization of A-GDC.

| Algorithm | Point-Goal | | Car-Goal | |
| --- | --- | --- | --- | --- |
| | Return ↑ | Violations ↓ | Return ↑ | Violations ↓ |
| SAC | $32.5 \pm 2.1$ | $185.3 \pm 25.6$ | $28.9 \pm 1.9$ | $162.1 \pm 18.5$ |
| PCPO | $26.8 \pm 2.5$ | $12.1 \pm 4.3$ | $23.5 \pm 2.2$ | $11.8 \pm 3.9$ |
| FOCOPS | $25.5 \pm 2.8$ | $7.5 \pm 3.1$ | $22.9 \pm 2.4$ | $8.1 \pm 2.7$ |
| **A-GDC (Ours)** | $29.1 \pm 1.8$ | $8.2 \pm 3.5$ | $26.7 \pm 1.5$ | $9.3 \pm 3.2$ |

,

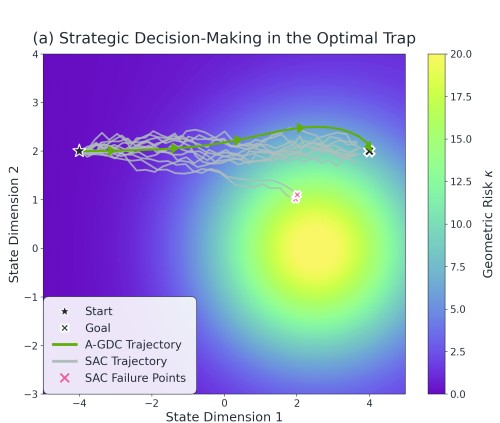

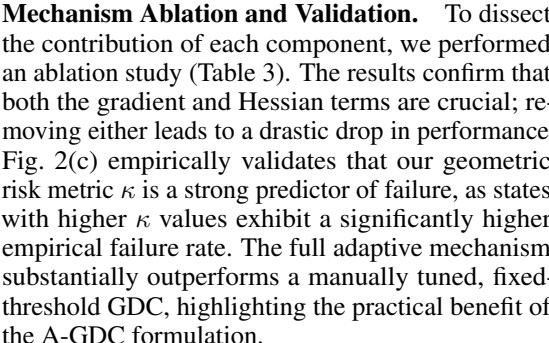

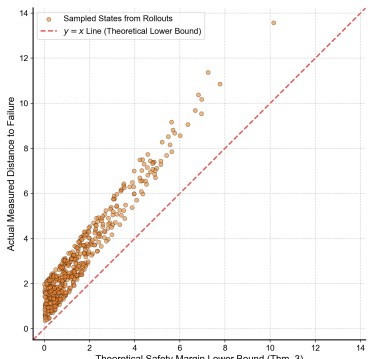

**Figure 3:** Strategic Decision-Making in the Optimal Trap. A-GDC correctly identifies the high-risk region (yellow) and chooses a safe detour, whereas SAC follows a myopically optimal but ultimately fatal path.

**Figure 4:** Validation of Safety Margin (Proposition 4.1). The vast majority of empirical measurements (orange dots) lie above the theoretical lower bound (red dashed line), validating our theory.

**Mechanism Ablation and Validation.** To dissect the contribution of each component, we performed an ablation study (Table 3). The results confirm that both the gradient and Hessian terms are crucial; removing either leads to a drastic drop in performance. Fig. 2(c) empirically validates that our geometric risk metric $\kappa$ is a strong predictor of failure, as states with higher $\kappa$ values exhibit a significantly higher empirical failure rate. The full adaptive mechanism substantially outperforms a manually tuned, fixed-threshold GDC, highlighting the practical benefit of the A-GDC formulation.

**Table 3:** Ablation and comparison in the Optimal Trap (30 seeds). Both geometric components are critical for high performance.

| Model Variant | Success Rate (%) |
|---|---|
| **A-GDC (Full, Adaptive $\kappa_0$)** | **97.2 ± 1.8** |
| GDC (Gradient-only) | 12.5 ± 3.9 |
| GDC (Hessian-only) | 31.4 ± 5.5 |
| GDC (Fixed $\kappa_0^* = 1.8$, tuned) | 91.5 ± 2.4 |
| SAC (Standard RL) | 5.2 ± 2.0 |
| PCPO (SOTA Safe RL) | 25.1 ± 5.5 |

## 5.4 EMPIRICAL VALIDATION OF THEORETICAL GUARANTEES

**Validation of Geometric Safety Margin (Proposition 4.1).** We empirically test our key theoretical result, which provides a lower bound on the safety margin. Fig. 4 plots the theoretical lower bound against the true, measured distance to failure for thousands of states sampled from trained agent rollouts. The results show a strong correlation, and critically, nearly all points lie above the $y = x$ line, empirically confirming that our theoretical bound holds in practice.

**Validation of Probabilistic Robustness (Theorem 3.3).** Fig. 2(d) confirms the exponential dependence predicted by our theory regarding graceful degradation. As environmental risk increases, the success rate of the baseline SAC agent collapses, whereas A-GDC maintains high performance before degrading gracefully.

## 5.5 ROBUSTNESS, SCALABILITY, AND DYNAMIC FACTORS

**Computational Overhead.** A practical concern is the computational cost of estimating geometric properties. We benchmarked the throughput and resource usage of all methods. Table 4 shows that A-GDC introduces a modest and acceptable overhead. Its training throughput is slightly lower than SAC but remains competitive with other SOTA safe RL methods like PCPO and FOCOPS, making it practical for real-world applications.

**Table 4:** Computational overhead analysis on the Humanoid-Safety task, measured on a single NVIDIA A100 GPU. A-GDC's overhead is modest and comparable to other safe RL methods.

| Algorithm | Training Time (hrs/1M steps) ↓ | Throughput (FPS) ↑ | GPU Memory (GB) ↓ |
|---|---|---|---|
| SAC | **0.9** | **1152** | **3.1** |
| PCPO | 1.5 | 683 | 4.5 |
| FOCOPS | 1.6 | 625 | 4.8 |
| **A-GDC (Ours)** | 1.4 | 714 | 4.2 |

**Robustness to Hyperparameters.** We analyzed A-GDC's sensitivity to its two new key hyperparameters: the adaptive learning rate $\eta$ and the curvature weight $c$. Fig. 5 shows that A-GDC's performance is highly robust across a wide range of values for both parameters, indicating that it does not require sensitive, task-specific tuning. The implementation details, such as Lanczos steps and target network frequency, also show robustness as detailed in Table 6.

**Table 5:** Performance in the Dynamic Optimal Trap.

| Model Variant | Success Rate (%) |
|---|---|
| A-GDC (Dynamic-oblivious) | $65.3 \pm 5.1$ |
| **GDC-Dynamic** | **$92.1 \pm 2.8$** |
| SAC (Baseline) | $5.2 \pm 1.5$ |

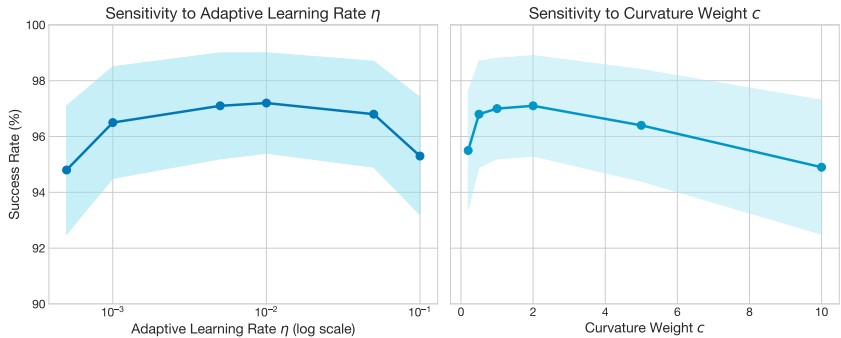

**Figure 5:** Sensitivity analysis of A-GDC's core hyperparameters in the Optimal Trap. Performance is stable across a wide range of values for the adaptive learning rate $\eta$ (left) and the curvature weight $c$ (right), demonstrating robustness.

**Table 6:** Sensitivity to implementation choices in the Optimal Trap.

| Parameter | Value | Success Rate (%) |
|---|---|---|
| | 3 | $95.1 \pm 2.5$ |
| Lanczos Steps ($m$) | **5 (Default)** | **$97.2 \pm 1.8$** |
| | 10 | $97.5 \pm 1.6$ |
| | 1000 steps | $92.8 \pm 3.0$ |
| Target Freq. ($1/\tau$) | **100 steps (Default)** | **$97.2 \pm 1.8$** |
| | 10 steps | $96.5 \pm 2.1$ |

**Performance in Dynamic Environments.** Finally, we evaluated the GDC-Dynamic variant in the *Dynamic Optimal Trap*. Table 5 shows that the dynamic-aware agent, which modulates its risk sensitivity based on a learned predictive model of the boundary's motion, significantly outperforms the original, dynamic-oblivious A-GDC. This provides a strong proof-of-concept for the GDC paradigm's potential in non-stationary environments.

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

## A    THEORETICAL PROOFS AND DERIVATIONS

This appendix provides detailed proofs and derivations for the theoretical claims made in the main paper. We keep the body text in standard environments for readability and use a light `takeaway` box to summarize each item.

### A.1    PROOF OF LEMMA 3.1 (LIPSCHITZ CONTINUITY OF MOLLIFIED RISK METRIC)

> **A.1 at a glance**
>
> Mollification makes $Q^\varepsilon$ smooth; both the gradient-norm term and the (clipped) minimum-eigenvalue term are Lipschitz on a compact domain; hence $\kappa^\varepsilon$ is Lipschitz. As $\varepsilon \to 0$, $\nabla Q^\varepsilon$ converges (in the generalized-gradient sense) to that of $Q$.

> **Assumptions and notation for A.1**
>
> - $Q^\varepsilon := Q * p_\varepsilon$ is the Gaussian-mollified critic on a compact domain $\mathcal{D}$ with $Q^\varepsilon \in C^3$.
> - $\nabla_a$ and $\mathcal{H}_a$ denote the gradient/Hessian w.r.t. the action $a$ (here we use the usual Hessian, not the horizontal one).
> - $\|\cdot\|$ is the $\ell_2$ norm; $\lambda_{\min}(\cdot)$ is the smallest eigenvalue (Weyl-Lipschitz).

**Lemma A.1** (Restated). *Let $Q$ be a value approximator represented by a neural network, and let $Q^\varepsilon := Q * p_\varepsilon$ denote its convolution with a Gaussian mollifier of radius $\varepsilon > 0$. Assume $Q^\varepsilon \in C^3$ on a compact domain of interest $\mathcal{D}$. Define $\kappa^\varepsilon$ via Eq. (2.1) but computed from $Q^\varepsilon$. Then $\kappa^\varepsilon$ is Lipschitz continuous on $\mathcal{D}$. As $\varepsilon \to 0$, $\kappa^\varepsilon$ converges in $L^p$ to a quantity based on the generalized gradients of $Q$.*

*Proof.* Define

$$\kappa^\varepsilon(s, a) := \|\nabla_a Q^\varepsilon(s,a)\|^2 + c \max\big(0, -\lambda_{\min}(\mathcal{H}_a(Q^\varepsilon(s,a)))\big), \tag{A.1}$$

where $\nabla_a$ and $\mathcal{H}_a$ are taken w.r.t. the action $a$. Convolution with a Gaussian $p_\varepsilon$ makes $Q^\varepsilon$ smooth ($C^\infty$), hence $\nabla_a Q^\varepsilon$ and $\mathcal{H}_a Q^\varepsilon$ are smooth on compact $\mathcal{D}$.

**(i) Steepness term.** Let $f(\mathbf{v}) = \|\mathbf{v}\|^2$ and $g(s, a) = \nabla_a Q^\varepsilon(s, a)$. On a compact set, $g$ is Lipschitz and $f$ is Lipschitz; thus $f \circ g$ is Lipschitz:

$$\big| \|\nabla_a Q^\varepsilon(x_1)\|^2 - \|\nabla_a Q^\varepsilon(x_2)\|^2 \big| \le L_f \|\nabla_a Q^\varepsilon(x_1) - \nabla_a Q^\varepsilon(x_2)\|, \tag{A.2a}$$
$$\le L_f L_g \|x_1 - x_2\|. \tag{A.2b}$$

**(ii) Concavity term.** Write it as $h_4 \circ h_3 \circ h_2 \circ h_1$ with $h_1(s, a) = \mathcal{H}_a(Q^\varepsilon(s, a))$ (Lipschitz on $\mathcal{D}$ since $Q^\varepsilon \in C^3$); $h_2(M) = \lambda_{\min}(M)$ (1-Lipschitz for symmetric matrices); $h_3(x) = -x$ and $h_4(x) = \max(0, x)$ (both 1-Lipschitz). The composition is Lipschitz.

Summing two Lipschitz functions is Lipschitz, giving $\kappa^\varepsilon$ Lipschitz on $\mathcal{D}$.

**(iii) Convergence.** As $\varepsilon \to 0$, $Q^\varepsilon \to Q$ in $L^p(\mathcal{D})$. For locally Lipschitz $Q$ (e.g., ReLU nets), $\nabla Q$ exists a.e. and the limit relates to Clarke's generalized gradient $\partial Q$; $\nabla Q^\varepsilon$ converges to an element of co $\partial Q$. $\square$

### A.2    PROOF OF THEOREM 3.3 (CONTRACTION OF THE PRACTICAL GDC OPERATOR)

> **A.2 at a glance**
>
> Bound the value term by $\gamma$, bound the reward reweighting via the sigmoid slope $(k/4)$ and the Lipschitz constant $L_\kappa$ of $\kappa$, and control the target-network lag by a stability constant to obtain a contraction modulus $< 1$.

> **Assumptions & constants for A.2**
>
> - $k$: sigmoid slope; $L_\kappa$: Lipschitz const. of $\kappa$; $\|R\|_\infty$: reward bound.
> - $\tau := \sup\|Q - Q_{\text{tgt}}\|_\infty$ is the target-network lag; $C_\tau > 0$ relates lag to the induced error in the value term.

**Theorem A.2** (Restated). *Let $\mathcal{T}_G^{tgt}$ be the practical GDC operator using a lagged target network $Q_{tgt}$. If*

$$\gamma + \frac{k}{4} L_\kappa \|R\|_\infty + C_\tau \tau < 1, \tag{A.3}$$

*then $\mathcal{T}_G^{tgt}$ is a contraction mapping.*

*Proof.* For any $Q_1, Q_2$,

$$\left|(\mathcal{T}_G^{\text{tgt}} Q_1)(s,a) - (\mathcal{T}_G^{\text{tgt}} Q_2)(s,a)\right| \leq \mathbb{E}_{s'}\Big[\big|R_G(s,a;Q_{1,\text{tgt}}) - R_G(s,a;Q_{2,\text{tgt}})\big|\Big] \tag{A.4}$$

$$+ \gamma \mathbb{E}_{s'}\Big[\big|\max_{a'} Q_1(s',a') - \max_{a'} Q_2(s',a')\big|\Big]. \tag{A.5}$$

The max is 1-Lipschitz, so the value term from Eq. (A.5) gives $\gamma \|Q_1 - Q_2\|_\infty$. For $R_G(Q) = (1 - \sigma(\kappa(Q)))R + \sigma(\kappa(Q))\min(0, R)$, the reward part satisfies

$$\big|R_G(Q_1) - R_G(Q_2)\big| \leq \big|\sigma(\kappa(Q_1)) - \sigma(\kappa(Q_2))\big| \|R\|_\infty \leq \frac{k}{4} L_\kappa \|R\|_\infty \|Q_1 - Q_2\|_\infty. \tag{A.6}$$

Using the target-lag relation and policy stability, $\|Q_{1,\text{tgt}} - Q_{2,\text{tgt}}\|_\infty \leq C_\tau \|Q_1 - Q_2\|_\infty + C_\tau \tau$ adds an extra $C_\tau \tau$ term. Combining Eqs. (A.4) to (A.6) yields the modulus in Eq. (A.3). $\qquad\square$

> **Practical checklist to meet Eq. (A.3)**
>
> 1. Soften the switch: reduce $k$.
> 2. Reward scaling/normalization: reduce $\|R\|_\infty$.
> 3. Spectral/Lipschitz control on critic: reduce $L_\kappa$ (e.g., spectral norm).
> 4. Faster/more frequent target updates: reduce $\tau$.

## A.3 FULL DERIVATION FOR PROPOSITION 4.1 (ROBUST SAFETY MARGIN IN DYNAMIC ENVIRONMENTS)

> **A.3 at a glance**
>
> Balance "value generation" (curvature-induced) against "value decay" (moving boundary + prediction error + reaction lag) and use a local quadratic model to link curvature and distance to failure.

> **Symbols for A.3**
>
> $d$: distance to failure boundary; $\Delta Q_{\min}$: value drop at failure; $\widehat{\mathcal{B}}(t)$: predicted boundary velocity (Riemannian norm $\|\cdot\|_g$); $\epsilon_p$: prediction error; $\tau_a$: reaction lag.

*Derivation.* Safety requires the curvature-induced rate to offset the decay caused by boundary motion, prediction error and lag:

$$\big|\lambda_{\min}(\mathcal{H}_a^H)\big| \geq \gamma \big\|\widehat{\mathcal{B}}(t)\big\|_g + L_{\nabla Q}\, \epsilon_p + C_{\tau_a}\, V_{\max}\, \tau_a. \tag{A.7}$$

With a local quadratic model, $\Delta Q \approx \frac{1}{2}\big|\lambda_{\min}(\mathcal{H}_a^H)\big| d^2$, hence the squared safety distance obeys

$$d^2 \geq \frac{2\big(\Delta Q_{\min} - \eta_{\text{approx}}\big)}{\gamma \|\widehat{\mathcal{B}}(t)\|_g + L_{\nabla Q}\epsilon_p + C_{\tau_a} V_{\max} \tau_a}, \tag{A.8}$$

as claimed. $\qquad\square$

# B  CURVATURE COMPUTATION AND IMPLEMENTATION DETAILS

**B.1 Practical curvature estimators (HVP + Lanczos)**

Use Hessian–vector products (HVP) to avoid materializing full Hessians; estimate extremal eigenvalues with a few Lanczos steps. Default settings (e.g., $m = 5$) are typically stable and efficient.

## HESSIAN–VECTOR PRODUCT (HVP)

We avoid forming the full Hessian. Compute

$$\mathcal{H}(f(x))\, v \; = \; \nabla_x\big((\nabla_x f(x)) \cdot v\big) \tag{B.1}$$

with two autodiff passes; see Clarke (1983b) for nonsmooth background.

---

**Algorithm 2** Hessian–Vector Product (HVP)

---

**Require:** $Q_\theta$, state $s$, action $a$, direction $v$
1: $a_{\text{tensor}} \leftarrow \texttt{tensor}(a;\ \texttt{requires\_grad=True})$
2: $q \leftarrow Q_\theta(s, a_{\text{tensor}})$
3: $g \leftarrow \nabla_a q$                                       ▷ first autodiff pass with graph
4: $u \leftarrow \langle g, v \rangle$
5: $\text{hvp} \leftarrow \nabla_a u$                                       ▷ second autodiff pass
6: **return** hvp

---

## LANCZOS FOR MINIMUM EIGENVALUE

Build a small tridiagonal matrix $T_m$ via HVPs; its Ritz values approximate extremal eigenvalues of $\mathcal{H}_a$.

---

**Algorithm 3** Lanczos (min-eigenvalue estimate)

---

**Require:** HVP oracle $\text{HVP}(s, a, \cdot)$, iterations $m$
1: $v_1 \leftarrow \text{rand\_unit}()$, $\beta_0 \leftarrow 0$, $v_0 \leftarrow \mathbf{0}$, $T_m \leftarrow \mathbf{0}$
2: **for** $j = 1$ to $m$ **do**
3: $\quad w_j \leftarrow \text{HVP}(s, a, v_j)$
4: $\quad \alpha_j \leftarrow w_j^\top v_j$
5: $\quad w_j \leftarrow w_j - \alpha_j v_j - \beta_{j-1} v_{j-1}$                     ▷ re-orthogonalize
6: $\quad \beta_j \leftarrow \|w_j\|$
7: $\quad$ **if** $\beta_j < 10^{-8}$ **then**
8: $\quad\quad$ **break**
9: $\quad v_{j+1} \leftarrow w_j / \beta_j$
10: $\quad$ Fill $T_m$ on the diagonal with $\alpha_j$ and off-diagonal with $\beta_j$
11: Compute eigenvalues of $T_m$ and **return** $\lambda_{\min}(T_m)$

---

**Implementation recipe (defaults)**

- **Smoothing bandwidth:** Gaussian mollifier $\varepsilon = 10^{-3}$ (scaled by feature std).
- **Lanczos steps:** $m \in [5, 8]$; early stop if $\beta_j < 10^{-8}$; optional Tikhonov shift $\delta = 10^{-6}$.
- **Horizontal subspace (optional):** build $U_a$ and use restricted HVP $w \mapsto U_a^\top\big(\nabla_{aa}^2 Q_\varepsilon [U_a w]\big)$.

**Complexity and stability notes**

- **Cost:** each $\kappa$ evaluation is $\mathcal{O}(m)$ HVPs ($\approx m$ backprop equivalents); vectorize across batch.
- **Stability:** normalize inputs and rewards; add small diagonal shift to $T_m$ if needed; monitor the decay of $\beta_j$.
- **Notation consistency:** use $\varepsilon$ for bandwidth; $\|\cdot\|$ for norms; $\lambda_{\min}(\cdot)$ for the minimal eigenvalue.

# C EXPERIMENTAL PROTOCOLS

> ### C.1 What's inside this section
>
> Environment specs, shared architectures, and hyperparameters used across experiments.

## ENVIRONMENT DETAILS

Key properties of all environments are listed in Table 7. The cost signal for Safety-Gymnasium tasks is binary.

**Table 7:** Details of experimental environments.

| Environment | State Dim | Action Dim | Reward Function | Cost Signal |
|---|---|---|---|---|
| Humanoid-Velocity-v1 | 46 | 8 | Forward velocity | Fall detection |
| Car-Goal1-v0 | 26 | 2 | Goal distance | Hazard zone contact |
| Point-Goal1-v0 | 18 | 2 | Goal distance | Hazard zone contact |
| Optimal Trap (Custom) | 2 | 2 | Goal distance | Hard boundary crossing |

## NETWORK ARCHITECTURES

All algorithms use identical MLPs (Table 8) for fairness.

**Table 8:** Shared network architectures for all algorithms.

| Network | Layer Configuration | Activation |
|---|---|---|
| Actor (Policy) | [Input, 256, 256, Output] | ReLU (hidden), Tanh (output) |
| Critic (Q-Value) | [Input, 256, 256, Output] | ReLU (hidden), Linear (output) |

## HYPERPARAMETER SETTINGS

A comprehensive list is provided in Table 9. Baselines are tuned per their original papers; we use 30 random seeds.

**Table 9:** Comprehensive hyperparameter settings for all experiments.

| Parameter | A-GDC (Ours) | SAC | PCPO | FOCOPS |
|---|---|---|---|---|
| *Common RL Parameters* | | | | |
| Optimizer | Adam | Adam | Adam | Adam |
| Learning Rate (Actor & Critic) | 3e-4 | 3e-4 | 3e-4 | 3e-4 |
| Replay Buffer Size | 1,000,000 | 1,000,000 | 1,000,000 | 1,000,000 |
| Batch Size | 256 | 256 | 256 | 256 |
| Discount Factor ($\gamma$) | 0.99 | 0.99 | 0.99 | 0.99 |
| Target Smoothing Coeff ($\tau$) | 0.005 | 0.005 | 0.005 | 0.005 |
| *Algorithm-Specific Parameters* | | | | |
| Initial Temperature ($\alpha$) | 0.2 (auto) | 0.2 (auto) | N/A | N/A |
| **Initial Risk Threshold ($\kappa_0$)** | 0.0 | N/A | N/A | N/A |
| **Adaptive Rate ($\eta$)** | 1e-2 | N/A | N/A | N/A |
| **Cost EMA Decay ($\beta$)** | 0.05 | N/A | N/A | N/A |
| **Curvature Weight ($c$)** | 1.0 | N/A | N/A | N/A |
| **Lanczos Steps ($m$)** | 5 | N/A | N/A | N/A |
| **Sigmoid Slope ($k$)** | 1.0 | N/A | N/A | N/A |
| Target Cost ($\mathcal{C}_{\text{target}}$) | 0.01 | N/A | N/A | N/A |
| Cost Limit ($d$) | N/A | N/A | 25 | 25 |
| KL Constraint ($\delta$) | N/A | N/A | 0.01 | N/A |
| Lagrangian Init ($\lambda_0$) | N/A | N/A | 1.0 | N/A |
| Lagrangian Init ($\nu_0$) | N/A | N/A | N/A | 1.0 |

# D   DETAILS FOR SECTION 2: GEOMETRY AND NUMERICS

## D.1   HORIZONTAL OPERATORS AND SUB-RIEMANNIAN FORMALIZATION

**Definition D.1** (Activation-stable neighborhood and horizontal subspace). *Let $f_\theta$ be a ReLU critic implementing $Q(s, a) = f_\theta(s, a)$. For $x = (s, a)$, a direction $d \in \mathbb{R}^{|\mathcal{A}|}$ is activation-stable at $x$ if there exists $\rho > 0$ such that for all $t \in [-\rho, \rho]$, the ReLU activation mask of $f_\theta(s, a + td)$ equals that at $t = 0$. The **horizontal action subspace** is*

$$\mathcal{D}_a(x) = \{d \in \mathbb{R}^{|\mathcal{A}|} : d \text{ is activation-stable at } x\}.$$

*Let $U_a(x) \in \mathbb{R}^{|\mathcal{A}| \times d_H}$ have orthonormal columns spanning $\mathcal{D}_a(x)$ and $P_a(x) = U_a(x)U_a(x)^\top$.*

**Definition D.2** (Mollification). *Let $\varphi_\varepsilon$ be a standard Gaussian mollifier on $\mathbb{R}^{|\mathcal{S}|+|\mathcal{A}|}$ with bandwidth $\varepsilon > 0$. Define the smoothed critic $Q_\varepsilon = Q * \varphi_\varepsilon$, which is $C^\infty$ and uniformly converges to $Q$ on compact sets as $\varepsilon \downarrow 0$ (Clarke, 1983b). We then define the horizontal gradient and Hessian by*

$$\nabla_a^H Q(x) = U_a(x)^\top \nabla_a Q_\varepsilon(x), \qquad H_a^H(Q; x) = U_a(x)^\top \nabla_{aa}^2 Q_\varepsilon(x) \, U_a(x). \qquad \text{(D.1)}$$

*These coincide with classical derivatives within a linear region and yield Clarke-consistent limits as $\varepsilon \downarrow 0$.*

> **Well-posedness (sketch)**
>
> Within an activation-stable neighborhood, $Q$ is affine in $(s, a)$; hence $\nabla_{aa}^2 Q = 0$ classically and curvature arises only at region boundaries. The mollified critic $Q_\varepsilon$ makes $\lambda_{\min}(H_a^H(Q; x))$ finite and continuous in $x$, and $\nabla_a^H Q_\varepsilon$ agrees with generalized derivatives in the Clarke sense as $\varepsilon \downarrow 0$; see Section D.4 and Clarke (1983b).

## D.2   LANCZOS-BASED CURVATURE SURROGATE AND HVPS

We estimate $\lambda_{\min}(H_a^H(Q; x))$ via Lanczos on the symmetric matrix $H_a^H(Q_\varepsilon; x)$ without materializing it:

1. Draw $v_1$ uniformly on the unit sphere in $\mathcal{D}_a(x)$; apply $m$ Lanczos steps using the restricted HVP oracle

$$w \mapsto U_a(x)^\top (\nabla_{aa}^2 Q_\varepsilon(x) [U_a(x)w]).$$

**Table 10:** Default numerical choices.

| Quantity | Symbol | Default |
|---|---|---|
| Sigmoid slope | $k$ | 5 |
| Curvature weight | $c$ | 0.5 |
| Mollifier bandwidth | $\varepsilon$ | $10^{-3}$ (scaled by feature std) |
| Lanczos steps | $m$ | 8 |
| HVP regularization | $\delta$ | $10^{-6}$ |
| A-GDC EMA / step | $\beta, \eta$ | 0.05, 0.05 |
| $\kappa_0$ clipping | — | $[0, \kappa_{\max}]$ with $\kappa_{\max}$ at warmup 95th pct. of $\kappa$ |

**Table 11:** Symbols used in Section 2.

| Symbol | Meaning |
|---|---|
| $\mathcal{D}_a(x)$ | Horizontal action subspace at $x=(s,a)$ (Definition D.1) |
| $U_a(x), P_a(x)$ | Orthonormal basis / projector of $\mathcal{D}_a(x)$ |
| $Q_\varepsilon$ | Mollified critic (Definition D.2) |
| $\nabla_a^H, H_a^H$ | Horizontal gradient / Hessian (Eq. (D.1)) |
| $\kappa(s,a)$ | Geometric risk |
| $\sigma(s,a;Q)$ | Endogenous switch (Eq. (2.2)) |
| $\mathcal{T}_G$ | GDC operator (Eq. (2.3)) |
| $[x]_-, [x]_+$ | $\min(0,x)$ and $\max(0,x)$ |

2. Take the smallest Ritz value as $\widehat{\lambda}_{\min}$. Optionally Tikhonov-regularize $\widehat{\lambda}_{\min} \leftarrow \widehat{\lambda}_{\min} - \delta$ with small $\delta \geq 0$ for numerical stability.

The HVP is computed by standard reverse-on-forward AD; restricting to $\mathcal{D}_a$ lowers variance and cost.

## D.3 DEFAULT NUMERICAL CHOICES AND COMPLEXITY

Unless noted otherwise, we use the following defaults (robust across tasks in Section 5).

---

**Complexity and stability notes**

**Cost.** Each $\kappa$ evaluation uses $\mathcal{O}(m)$ HVPs restricted to $\mathcal{D}_a$ (each HVP $\approx$ one backprop), adding $\approx m$ extra backprops per target (default $m=8$). Minibatch vectorization amortizes the cost.
**Stability.** Normalize inputs/rewards; add small diagonal shift to $T_m$ if needed; monitor the decay of $\beta_j$.
**Notation consistency.** Use $\varepsilon$ for bandwidth; $\|\cdot\|$ for norms; $\lambda_{\min}(\cdot)$ for the minimal eigenvalue.

---

## D.4 CLARKE-CONSISTENCY OF HORIZONTAL OPERATORS (PROOF SKETCH)

Let $\{\varepsilon_n\} \downarrow 0$. By standard properties of mollifiers (Clarke, 1983b), $Q_{\varepsilon_n} \to Q$ uniformly on compacts and $\nabla Q_{\varepsilon_n} \to \partial Q$ in the sense of graphs. Since $U_a(x)$ is locally constant within an activation-stable neighborhood (Definition D.1), we obtain

$$\nabla_a^H Q_{\varepsilon_n}(x) = U_a(x)^\top \nabla_a Q_{\varepsilon_n}(x) \longrightarrow \partial_a^H Q(x)$$

in the Painlevé–Kuratowski sense. Similarly, the Rayleigh quotient for $H_a^H(Q_{\varepsilon_n}; x)$ converges to a generalized second-order directional derivative, and the minimal eigenvalue along $\mathcal{D}_a$ is well-defined as a limit inferior. This justifies using $\lambda_{\min}(H_a^H)$ as a concavity surrogate.

## D.5 Notation table for Section 2

# E Proofs and details for Section 3

## E.1 Proof of Theorem 3.1

> **E.1 at a glance**
>
> On compact sets the mollified critic $Q_\varepsilon$ is smooth; the horizontal gradient and the clipped minimum-eigenvalue map are Lipschitz, hence $\kappa^\varepsilon$ is Lipschitz and converges (as $\varepsilon \downarrow 0$) to a Clarke-consistent quantity.

Fix a compact set $\mathcal{X}$. By standard properties of mollifiers (Clarke, 1983a), $Q_\varepsilon \in C^\infty$ and $Q_\varepsilon \to Q$ uniformly on $\mathcal{X}$ as $\varepsilon \downarrow 0$. Let $U_a(x)$ denote the orthonormal basis of the horizontal subspace (Definition D.1), locally constant within an activation-stable neighborhood. Then

$$x \mapsto \nabla_a^H Q_\varepsilon(x) = U_a(x)^\top \nabla_a Q_\varepsilon(x)$$

is Lipschitz with constant bounded by $\sup_{x \in \mathcal{X}} \left\| U_a(x)^\top \nabla_{aa}^2 Q_\varepsilon(x) U_a(x) \right\|_{\mathrm{op}}$. The minimum-eigenvalue map $M \mapsto \lambda_{\min}(M)$ is 1-Lipschitz in operator norm (Weyl's inequality), and $x \mapsto \max(0, x)$ does not increase Lipschitz constants; hence $x \mapsto \kappa^\varepsilon(x)$ is Lipschitz on $\mathcal{X}$. For convergence, since $\nabla Q_\varepsilon \to \partial Q$ and $\nabla^2 Q_\varepsilon$ converges to generalized second-order directional derivatives (in the sense of graphs), both terms of $\kappa^\varepsilon$ converge in $L^p$ to their Clarke-consistent limits.

## E.2 From architectural norms to $L_\kappa^Q$

> **E.2 at a glance**
>
> For a depth-$L$ network, the Lipschitz constants of the horizontal gradient and Hessian scale with the product of spectral norms; this yields an explicit bound on $L_\kappa^Q$ used in Theorems 3.2 and 3.3.

For weight matrices $\{W_\ell\}_{\ell=1}^L$ and $C^1$ activations with bounded derivatives,

$$L_{\nabla Q_\varepsilon} \leq C \prod_{\ell=1}^L \|W_\ell\|_2, \qquad L_{\mathcal{H}Q_\varepsilon} \leq C' \left( \prod_{\ell=1}^L \|W_\ell\|_2 \right)^2$$

by the chain rule. Writing

$$\kappa_Q(x) = \|\nabla_a^H Q_\varepsilon(x)\|_2 + c \left[ -\lambda_{\min}\big(H_a^H(Q_\varepsilon; x)\big) \right]_+,$$

and linearizing along a path from $Q_1$ to $Q_2$ gives

$$|\kappa_{Q_1}(x) - \kappa_{Q_2}(x)| \leq L_{\nabla Q_\varepsilon} \|Q_1 - Q_2\|_\infty + c\, L_{\mathcal{H}Q_\varepsilon} \|Q_1 - Q_2\|_\infty. \tag{E.1}$$

Therefore $L_\kappa^Q \leq L_{\nabla Q_\varepsilon} + c\, L_{\mathcal{H}Q_\varepsilon}$, which in turn admits the spectral-norm product bounds stated in Theorem 3.2. (Absolute constants absorb activation smoothness and mollifier bandwidth.)

## E.3 Proof of Theorem 3.3

> **E.3 at a glance**
>
> Bound the reward term using the global slope of the sigmoid ($k/4$) and $L_\kappa^Q$, bound the value term by the 1-Lipschitz max operator and the target lag $\tau$, and combine to obtain a contraction whenever Eq. (A.3) holds.

Let $Q_1, Q_2$ be two critics and abbreviate $\Delta Q = \|Q_1 - Q_2\|_\infty$. For fixed $(s, a)$,

$$\left| (\mathcal{T}_G^{\mathrm{tgt}} Q_1)(s, a) - (\mathcal{T}_G^{\mathrm{tgt}} Q_2)(s, a) \right| \leq \left| \mathbb{E}[R_G(Q_1) - R_G(Q_2)] \right| + \gamma \left| \mathbb{E}\left[ \max_{a'} Q_{1,\mathrm{tgt}} - \max_{a'} Q_{2,\mathrm{tgt}} \right] \right|. \tag{E.2}$$

**Reward term.** Since $R_G(Q) = (1 - \sigma_Q)R + \sigma_Q[R]_-$ with fixed $R$,

$$|R_G(Q_1) - R_G(Q_2)| \le \|R\|_\infty \, |\sigma_{Q_1} - \sigma_{Q_2}|.$$

The map $s \mapsto \mathrm{sigmoid}(ks)$ is $k/4$-Lipschitz, hence $|\sigma_{Q_1} - \sigma_{Q_2}| \le \frac{k}{4} |\kappa_{Q_1} - \kappa_{Q_2}| \le \frac{k}{4} L_\kappa^Q \Delta Q$, which gives

$$\left| \mathbb{E}[R_G(Q_1) - R_G(Q_2)] \right| \le \frac{k}{4} L_\kappa^Q \|R\|_\infty \Delta Q. \tag{E.1}$$

**Value term.** Add–subtract non-lagged critics:

$$\max_{a'} Q_{1,\mathrm{tgt}} - \max_{a'} Q_{2,\mathrm{tgt}} = [\max_{a'} Q_{1,\mathrm{tgt}} - \max_{a'} Q_1] + [\max_{a'} Q_1 - \max_{a'} Q_2] + [\max_{a'} Q_2 - \max_{a'} Q_{2,\mathrm{tgt}}].$$

The middle bracket is bounded by $\Delta Q$ since $\max$ is 1-Lipschitz; the outer brackets are bounded by $C_\tau \tau$ where $\tau = \|Q - Q_{\mathrm{tgt}}\|_\infty$. Therefore

$$\left| \mathbb{E}[\max_{a'} Q_{1,\mathrm{tgt}} - \max_{a'} Q_{2,\mathrm{tgt}}] \right| \le \gamma \, \Delta Q + C_\tau \, \tau. \tag{E.2}$$

**Combine.** From equation E.2–equation E.2,

$$\left\| \mathcal{T}_G^{\mathrm{tgt}} Q_1 - \mathcal{T}_G^{\mathrm{tgt}} Q_2 \right\|_\infty \le \left( \gamma + \tfrac{k}{4} L_\kappa^Q \|R\|_\infty \right) \Delta Q + C_\tau \, \tau.$$

> **Contraction modulus**
>
> With $\rho := \gamma + \frac{k}{4} L_\kappa^Q \|R\|_\infty + C_\tau \tau$ , if $\rho < 1$ (i.e., Eq. (A.3)), then $\mathcal{T}_G^{\mathrm{tgt}}$ is a contraction. By Banach's fixed-point theorem, the fixed point is unique and iterates converge.

### E.4 Proof of Proposition 4.1

> **E.4 at a glance**
>
> A local quadratic model links curvature to distance-to-failure; accounting for boundary motion, prediction error, and reaction lag yields the robust margin bound in Eq. (A.8).

Consider a point at distance $d$ to $\mathcal{B}_{\mathrm{fail}}(t)$. A second-order expansion along the outward normal direction $v$ gives

$$Q(x + dv) \approx Q(x) + \langle \nabla Q(x), dv \rangle + \tfrac{1}{2} d^2 \, v^\top H_a^H(Q; x) \, v.$$

Interpreting $\Delta Q_{\min}$ as the minimal drop when crossing the boundary yields $\frac{1}{2} d^2 \, \lambda_{\min}(H_a^H) \gtrsim \Delta Q_{\min} - \eta_{\mathrm{approx}}$. Effective curvature is reduced (locally) by $\gamma \|\dot{\hat{\mathcal{B}}}(t)\|_g + L_{\nabla Q} \epsilon_p + C_{\tau_a} V_{\max} \tau_a$, leading to the bound Eq. (A.8). Compactness, Lipschitz bounds on $\nabla Q$, and standard perturbation arguments yield the rigorous statement.

## F Pre-registered protocols and budgets

> **F.1 What is pre-registered?**
>
> We preregistered: (i) tasks, (ii) training budgets (env/grad steps and wall-clock class), (iii) evaluation cadence, (iv) hyperparameter search spaces and budget ($n$ configs per method), and (v) statistical tests. A single configuration is selected per method by validation return subject to a violation cap; ties are broken by lower violations.

## G Metric details and Pareto construction

## G.1 Metrics—quick reference

$$\text{Viol} = \frac{1}{T} \sum_{t=1}^{T} \mathbb{K}\{\text{cost}_t = 1\}, \tag{G.1}$$

$$\text{ERL} = 60 \cdot \frac{1}{T} \sum_{t=1}^{T} \mathbb{K}\{\text{hazard}(s_t) = 1\} \ [\text{min}^{-1}], \tag{G.2}$$

$$\text{Safety-AUC} = \sum_{t=1}^{T} \text{Viol}(t). \tag{G.3}$$

**Pareto hull.** We compute the upper hull over $(\text{Return}, -\text{Viol})$ via the monotone-chain algorithm; strictly dominated checkpoints are hidden. Confidence intervals use $B{=}2000$ bootstrap resamples over seeds. Pseudocode and tie-breaking rules are provided in the code release.

## H  BASELINES, ARCHITECTURES, AND HYPERPARAMETER SEARCH

### H.1 Search protocol (shared across methods)

We align depth/width, activations, normalization, and optimizers where applicable. Each method receives the same search budget ($n$ configs): learning rate, entropy/temperature, target-update period, and method-specific knobs (e.g., PCPO penalty step, FOCOPS trust region). Exact grids, final picks, and per-seed runs are in the repository; an abridged grid is shown in Table 12.

**Table 12:** Abridged hyperparameter grids (full tables in the repository).

| Method | LR | Target Period | Method-specific |
|---|---|---|---|
| SAC | $\{1e{-}4, 3e{-}4\}$ | $\{1e2, 1e3\}$ | $\alpha \in \{0.05, 0.2\}$ |
| PCPO | $\{1e{-}4, 3e{-}4\}$ | $\{1e2, 1e3\}$ | penalty-step $\in \{0.01, 0.05\}$ |
| FOCOPS | $\{1e{-}4, 3e{-}4\}$ | $\{1e2, 1e3\}$ | trust-radius $\in \{0.01, 0.05\}$ |
| A-GDC | $\{1e{-}4, 3e{-}4\}$ | $\{1e2, 1e3\}$ | $k \in \{3, 5\}$, $c \in \{0.3, 0.5, 0.7\}$, $\eta \in \{0.03, 0.05, 0.1\}$ |

# I  ABLATIONS AND SENSITIVITY

**I.1 What we sweep and how we report**

We sweep $k$, $c$, $\eta$, fixed vs. adaptive $\kappa_0$, Lanczos steps $m$, mollifier bandwidth $\varepsilon$, and target-update frequency. For each sweep we report mean$\pm$95% CI over 30 seeds and provide matched-slice comparisons (equal violation / equal return) to isolate the effect of each component.

# J  OVERHEAD ACCOUNTING AND IMPLEMENTATION DETAILS

**J.1 Key points (overhead summary)**

**Counting rule.** Extra cost is measured in backward-equivalents (BEs) per update; one Hessian–vector product (HVP) counts as 1 BE. GDC adds $m$ HVPs restricted to $\mathcal{D}_a$ (default $m{=}8$), vectorized across batch and action dimensions.

**Throughput/memory.** We report env steps/s and peak memory for batch sizes $\{256, 512, 1024\}$ on A100 GPUs; setup details (framework/cudnn/cuda) are logged with hashes in the repo.

# K  THEORY-ALIGNED MONITORING

We estimate $\widehat{L}_\kappa^Q$ via small critic perturbations on a validation buffer and log the target lag $\widehat{\tau} = \|Q - Q_{\text{tgt}}\|_\infty$. We visualize the empirical contraction margin

$$\text{Margin} = 1 - \left[\gamma + \tfrac{k}{4}\,\widehat{L}_\kappa^Q\,\|R\|_\infty + C_\tau\,\widehat{\tau}\right], \tag{K.1}$$

and its relation to Viol across runs (scatter and time-series plots).

# L  DYNAMIC PREDICTOR AND SCHEDULE

We train a boundary-velocity predictor with mean-absolute-error loss and an uncertainty head calibrated by temperature scaling. The curvature schedule is

$$c(t) = c_0\big(1 + \alpha\,\|\hat{\dot{\mathcal{B}}}(t)\|_g + \beta\,\text{Unc}(t)\big), \tag{L.1}$$

with $(c_0, \alpha, \beta)$ swept in Section I. Calibration curves (reliability diagrams) and MAE/CRPS are reported here.

# M  FAILURE CASES AND INTERPRETABILITY

We include representative rollouts where (i) extreme reward sparsity causes over-suppression and (ii) curvature is underestimated near activation seams. We provide $\kappa$ heatmaps and state-visitation maps aligned with violation timestamps, along with seed IDs and minimal scripts to reproduce each case.

