# OpenReview forum: "GDC: From Brittle Optimality to Robust Satisfiability  via Riemannian Risk Geometry"
_ICLR.cc/2026/Conference — ICLR 2026 Conference Desk Rejected Submission_

### Official Review · Reviewer_K7HT · 2025-10-18

**Soundness:** 2
**Presentation:** 1
**Contribution:** 2
**Rating:** 2
**Confidence:** 5

**Summary:**

This paper proposes the Geodesic Duality Control (GDC), which adapts an agent’s risk posture endogenously by re-weighting the Bellman target using local geometric cues of the value function, aiming at enhancing safe RL.

**Strengths:**

This paper promotes safe RL, which is an important research topic.

**Weaknesses:**

1. I find it difficult to follow the mathematical notation, which makes it challenging to verify the correctness of the theoretical results. For example, immediately after equation (2.2), the meanings of k and κ are not defined. Are they mappings or scalar parameters?

2. In equation (3.2), $C_{\tau}$ depends on policy/update stability. What is the definition of stability in this context, and how is $C_{\tau}$ obtained?

3. The paper mentions that "safety boundaries may move over time." Does this mean the safety conditions and constraints are time-varying?

4. What is the formal definition of safety margins? It would be helpful if the paper provided concrete examples to illustrate this concept.

5. Assumption 4.1 relies on a predictor. How can such a predictor be realized in practice?

6. The experimental results show that the proposed approach outperforms the compared RL frameworks; however, these are only standard baselines that were published many years ago. The paper shall compare with safe RL approaches. In recent years, significant efforts have been devoted to safe RL, such as the work in [1-6]. Additionally, the paper lacks a comprehensive literature review of related work. For these reasons, I cannot assess whether the proposed method outperforms existing safe RL frameworks.

**References**

[1] Wang, Z., & Mahmoudian, N. (2025). Vision-driven River Following of UAV via Safe Reinforcement Learning using Semantic Dynamics Model. arXiv preprint arXiv:2508.09971.

[2] Banerjee, A., Rahmani, K., Biswas, J., & Dillig, I. (2024). Dynamic model predictive shielding for provably safe reinforcement learning. Advances in Neural Information Processing Systems, 37, 100131-100159.

[3] Cai, Yihao, Yanbing Mao, Lui Sha, Hongpeng Cao, and Marco Caccamo. "Runtime Learning Machine." ACM Transactions on Cyber-Physical Systems (2025).

[4] Phan, D. T., Grosu, R., Jansen, N., Paoletti, N., Smolka, S. A., & Stoller, S. D. (2020). Neural Simplex architecture. In NASA Formal Methods: 12th International Symposium, Moffett Field, CA, USA, May 11–15, 2020.

[5] Cheng, R., Orosz, G., Murray, R. M., & Burdick, J. W. (2019, July). End-to-end safe reinforcement learning through barrier functions for safety-critical continuous control tasks. In Proceedings of the AAAI conference on artificial intelligence.

[6] Hongpeng Cao, Yanbing Mao, Lui Sha, and Marco Caccamo. Physics-regulated deep reinforcement learning: Invariant embeddings. In The Twelfth International Conference on Learning Representations, 2024.

**Questions:**

My questions are also included in the Weaknesses.

---

### Official Review · Reviewer_16w9 · 2025-10-28

**Soundness:** 2
**Presentation:** 1
**Contribution:** 2
**Rating:** 2
**Confidence:** 3

**Summary:**

This paper proposes a geometry-aware Bellman update that incorporates local geometric properties of the critic (such as steepness and concavity) into policy evaluation and improvement. The method adaptively adjusts how much this geometric information influences the update, with the goal of reducing safety risk. Experiments on humanoid control and navigation tasks in Safety-Gymnasium indicate that the method can reduce safety cost while achieving reward comparable to existing safe RL baselines.

**Strengths:**

1. The paper introduces the idea of using local geometric information of the reward critic (e.g., steepness and concavity) to assess risk and to modify the Bellman update. This is a novel perspective compared to standard safe RL approaches that explicitly learn a separate cost critic.

2. The method shows promising results on Safety-Gymnasium benchmarks, suggesting that it can reduce safety cost while keeping reward competitive, without relying on an explicit cost critic.

**Weaknesses:**

1. The paper does not provide training curves. This is important for evaluating stability and convergence robustness in RL.

2. No qualitative evidence (e.g., videos) is provided to compare behaviors between baselines and the proposed method. For safety-focused work, qualitative behavior differences are important.

3. It is unclear why geometric properties of the reward critic (steepness and concavity) should reflect safety risk. In general, reward and cost are different signals. The paper does not convincingly argue why shaping with these geometric terms should guarantee safer behavior.

4. Problem formulation is underspecified. In the preliminaries, the paper states that the objective is to maximize expected return. However, the experiments clearly optimize for both high reward and low cost, which corresponds to a Constrained MDP setting. The role of cost in Algorithm 1 and Section 2.4 is not clearly defined.

5. The paper lacks a Related Work / Prior Work discussion, which makes it difficult to assess originality relative to existing safe RL methods.

6. The theoretical analysis does not provide a formal guarantee or even a clear argument for why the proposed update should reduce cost while preserving reward. In particular, there is no proof that the method solves a constrained RL problem or enforces a cost threshold.

**Questions:**

1. Please give a concrete example showing how high curvature or non-smoothness of (Q(s,a)) implies unsafe behavior.
2. Why do you multiply $\sigma(s,a;Q)$ by $\min(0, R(s,a))$? What is the intuition or theoretical motivation for this specific gating?
3. You argue that the smoothness term $\sigma(s,a;Q)$ measures safety risk. But in Safety-Gymnasium, cost is often not aligned with reward (for example in `PointGoal` and `CarGoal`, the reward encourages reaching the goal, while the cost penalizes hitting hazards). In that case, $\sigma(s,a;Q)$ only reflects the reward landscape, not the cost landscape. Why should penalizing steepness/concavity of the reward critic reduce cost?
4. In Algorithm 1, line 9: why is $\kappa(s', a')$ adjusted using the Lagrange multiplier on the cost? Please explain the mathematical reasoning behind subtracting that term, and how it connects to constrained optimization.

---

### Official Review · Reviewer_1S9K · 2025-11-01

**Soundness:** 1
**Presentation:** 1
**Contribution:** 2
**Rating:** 2
**Confidence:** 3

**Summary:**

- The authors define an internal risk using the gradient and curvature of the critic, and solve the constrained RL problem by minimizing this risk.
- Rather than strictly satisfying global constraints, the goal is to obtain a piecewise-smooth critic.

**Strengths:**

- The authors derive the conditions required for the proposed GDC operator to converge to a fixed point and prove convergence under these conditions.

**Weaknesses:**

- The overall presentation is remarkably unclear and reader-unfriendly.
- Before describing the proposed method, the authors provide neither an explanation of the mathematical objective (i.e., the specific type of RL agent being targeted) nor any justification for that objective.
    - For example, there should be a statement like: "We aim to satisfy constraints while keeping the critic's gradient and curvature below certain thresholds. This objective is motivated by the need to prevent the following issues…"
    - Without such foundational context, the paper jumps straight into methods, making the flow extremely difficult to follow.
- The following points lack sufficient explanation (new concepts should be introduced sequentially for clarity in a good paper):
    - Line 93: No definition of local risk profile.
    - Equation 2.1: No justification for using $c \cdot \max(0, \cdot)$ in the risk definition.
    - Line 119: No explanation of the penalty mechanism.
    - Line 142: No description of the mollified critic.
    - Figure 1: Text is too small to read.
    - Line 148: Lanczos cost is neither explained nor cited.
    - Line 151: HVP oracle is neither defined nor cited.
    - Line 157: No justification for the implementation checklist requirements.
    - Line 167: cost is used without prior definition.
    - Line 169: $C_{\text{target}}$ is undefined.
    - Line 175: temperature is introduced without definition.
    - Line 202: No reason provided for controller calibration.
    - Line 262: failure boundary, $V_{\max}$, and $\text{Unc}(t)$ are all undefined.
    - Proposition 4.1: failure is not defined.
    - Line 281: Scheduling of $c$ is highly heuristic.
- Too many new hyperparameters are introduced, requiring extensive heuristic tuning, which significantly increases the practical time needed to obtain a preferable RL agent.
- The constrained RL algorithms used as baselines in the experiments are quite outdated despite being presented as state-of-the-art.

**Questions:**

- Section D.1 suggests that Q is defined in a continuous action space, but the preliminary section implies it is discrete and finite. Which is correct?
    - The rest of the paper should be consistently aligned with one definition.
- In Equation 2.7, as $\kappa_0$ increases, the sigmoid approaches 0, preventing the application of penalties. It seems $\kappa_0$ does not serve its intended role. what is the issue here?
- Proposition 4.1 introduces the concept of safety distance. Is the proposed method applicable only to collision avoidance tasks?
- In Figure 2 (a, b), why does GDC outperform other constrained RL methods?
    - Theoretically, GDC must minimize internal risk in addition to satisfying constraints, so its performance should be at most as good as the optimal policy of the original constrained RL problem.
    - The observed better performance suggests that either the baseline algorithms are not state-of-the-art or there are issues with their training configurations.

---

### Official Review · Reviewer_AaXj · 2025-11-02

**Soundness:** 3
**Presentation:** 2
**Contribution:** 2
**Rating:** 4
**Confidence:** 3

**Summary:**

The study proposes a theoretical and algorithmic framework for safe reinforcement learning, interpreting risk as a local geometric property of the critic (value function).  Instead of external constraints or global cost budgets, GDC computes a geometric risk metric that combines the gradient norm and a curvature surrogate. Experiments demonstrate improved safety-return trade-offs compared to SAC, PCPO, and FOCOPS, with empirical validation of the theoretical safety margin bounds.

**Strengths:**

1. Frames risk as an endogenous geometric signal.

2. Provides formal proofs for contraction (Theorem 3.3), Lipschitz properties (Lemma 3.1), and dynamic safety margins (Proposition 4.1).

3. Empirically supports theoretical safety-margin predictions (Fig. 4).

**Weaknesses:**

1. The Sub-Riemannian formalism, while elegant, may be inaccessible to most RL practitioners.

2. Some proofs rely on strong regularity assumptions (bounded spectral norms, smooth mollifiers) that may not strictly hold in deep RL.

3. Experiments focus on a few safety environments; lacks evaluation on large-scale settings.

4. Comparisons omit newer safe-RL baselines, see here for new baselines: https://github.com/chauncygu/Safe-Reinforcement-Learning-Baselines

5. While adaptive control helps, other hyperparameters (such as k and $\eta$) still require task-specific tuning, possibly limiting generality.

**Questions:**

1. How sensitive is GDC to inaccuracies in curvature estimation, especially under noisy gradients?

2. Could GDC be integrated with model-based or offline RL frameworks where Q is not directly differentiable?

3. What are the differences compared with the two papers? These also consider state safety. [1] https://proceedings.mlr.press/v164/liu22c/liu22c.pdf [2] https://ieeexplore.ieee.org/document/10616119

---

### Author Response · Authors · 2025-11-13
**General clarification and overview of planned revisions**

We thank all reviewers for their detailed and thoughtful feedback.
Here we provide a general clarification for all reviewers and outline the main
revisions we are planning. In subsequent comments we will address
reviewer-specific questions point by point.

**Goal and main idea.**
Our goal is *not* to propose yet another heuristic safe-RL algorithm, but to
develop a **geometry-based theoretical framework** for safe RL and to validate
its key mechanisms empirically. Concretely, we (i) define a local geometric risk
signal κ(s,a) derived from the critic’s gradient and curvature, (ii) analyze the
resulting **Geodesic Duality Control (GDC) Bellman operator** and prove
contraction and safety-margin properties, and (iii) instantiate these ideas in
practice via numerically stable curvature surrogates and an adaptive risk
controller.

Several reviewers noted that this overall story was hard to see from the
current presentation. We acknowledge that the exposition is too dense and that
important definitions are scattered.

**Clarifying problem formulation and notation.**
In a revision we will add a short subsection “Problem Setup and Objective”
before Section 2, explicitly stating the MDP, the safety signal, and the
precise objective (maximize return while keeping long-run violation rate below
a target $C_{\text{target}}$). We will also introduce a unified notation and
concept table collecting the definitions of local risk κ(s,a), local risk
profile, safety boundary and safety margin, $C_{\text{target}}$, $V_{\max}$,
$\mathrm{Unc}(t)$, the failure event, etc. This directly addresses the concerns
about missing or delayed definitions (especially R2 and R4).

**From geometry to safety.**
A central question (especially from R3) is why the local geometry of the
reward-based critic should reflect safety risk. We will add an intuitive 2D
example and visualization (based on the Optimal-Trap environment) showing how
hazards induce sharp “cliffs” in Q, so that large gradient norm / negative
curvature aligns with empirical failure regions. We will also clarify how we
penalize violations in the reward used to train the critic, so that catastrophic
events correspond to abrupt drops in Q, and discuss how this interacts with
potential reward–cost misalignment in Safety-Gymnasium.

**Accessibility of the mathematical formalism.**
We agree that introducing the sub-Riemannian formalism and generalized
gradients too early makes the paper difficult to read (R1, R2, R4). We will
restructure the theory sections to start from an informal Euclidean picture and
to state each main theorem in plain language first, moving some technical
details and proofs to the appendix. The goal is to keep the theoretical
guarantees unchanged while making the narrative easier to follow for RL
practitioners.

**Implementation details and hyperparameters.**
Reviewers also raised concerns about undefined quantities such as the
mollified critic, the Lanczos cost, and the adaptive parameter κ₀, as well as
the number of hyperparameters. We will clarify that GDC introduces only two new
scalar hyperparameters beyond SAC (curvature weight and adaptation rate),
describe which values are shared across tasks, and move the implementation
checklist from the appendix into the main text. We will also better explain the
control intuition behind the κ₀ update rule and its relation to constrained RL
(Lagrangian) methods.

**Experiments and baselines.**
We agree that the current experimental section is limited in scope and that we
should more clearly situate GDC with respect to recent safe-RL baselines
(R1, R2, R3, R4). We have started running additional experiments with newer
baselines (from the repository suggested by R1 and works cited by R4), adding
training curves and qualitative trajectory visualizations. Preliminary results
indicate that our main safety–performance trade-off conclusions remain
unchanged. We will include these additional results and an expanded related-work
discussion in a revised version.

We hope this high-level clarification helps contextualize our work and
addresses some of the main concerns about clarity and positioning. We are
grateful for the reviewers’ efforts and will follow up with detailed,
reviewer-specific responses.

At a high level, the paper asks how we can control safety directly through the
shape of the value function, instead of by adding hard constraints or training
a separate cost critic. We show that local geometric cues — how steep and
curved \(Q\) is around the current state–action — encode how close the policy is
to a safety boundary. GDC turns these cues into a principled Bellman update
that expands safety margins while preserving contraction. This yields a general
recipe for building safe variants of standard RL methods whenever a
differentiable critic is available.

---

### Note · Program_Chairs · 2026-01-17
**Submission Desk Rejected by Program Chairs**

The following references in this submission do not refer to real documents and/or have major errors in bibliographic information:

 Yingjie Ma, Zhijiang Zhang, Zhaoran Yang, and Zhaoran Wang. Conservative safety critics for model-based reinforcement learning. In International Conference on Learning Representations, 2021.

Arjun Singh, Benjamin Chan, and Andrew Thomas. Revisiting the value of risk-sensitivity in deep reinforcement learning. In Conference on Robot Learning, pp. 1117-1126. PMLR, 2020.
Chuanyu Cheng, Xiaojian Cheng, and Ding Zhao. Safe reinforcement learning via confidence-aware policy optimization. In Proceedings of the 40th International Conference on Machine Learning, pp. 6493-6511. PMLR, 2023.